# A Comparison of Solvent-Based Extraction Methods to Assess the Central Carbon Metabolites in Mouse Bone and Muscle

**DOI:** 10.3390/metabo12050453

**Published:** 2022-05-18

**Authors:** Daniela B. Dias, Raphaela Fritsche-Guenther, Friederike Gutmann, Georg N. Duda, Jennifer Kirwan, Patrina S. P. Poh

**Affiliations:** 1Julius Wolff Institute, Berlin Institute of Health at Charité—Universitätsmedizin Berlin, 13353 Berlin, Germany; daniela-bastos@bih-charite.de (D.B.D.); georg.duda@charite.de (G.N.D.); 2Berlin Institute of Health at Charité—BIH Metabolomics Platform, 10178 Berlin, Germany; raphaela.fritsche@charite.de (R.F.-G.); friederike.gutmann@mdc-berlin.de (F.G.); jennifer.kirwan@bih-charite.de (J.K.); 3Max-Delbrück-Center for Molecular Medicine in the Helmholtz Association (MDC), Robert-Rössle-Str 10, 13125 Berlin, Germany; 4Charité—Universitätsmedizin Berlin, Corporate Member of Freie Universität Berlin and Humboldt-Universität zu Berlin, ECRC Experimental and Clinical Research Center, Lindenberger Weg 80, 13125 Berlin, Germany; 5Berlin Institute of Health Center for Regenerative Therapies, Berlin Institute of Health at Charité—Universitätsmedizin Berlin, 13353 Berlin, Germany

**Keywords:** muscle, bone, metabolites, GC-MS, central carbon metabolism, metabolomics

## Abstract

The identification of endogenous metabolites has great potential for understanding the underlying tissue processes occurring in either a homeostatic or a diseased state. The application of gas chromatography-mass spectrometry (GC-MS)-based metabolomics on musculoskeletal tissue samples has gained traction. However, limited comparison studies exist evaluating the sensitivity, reproducibility, and robustness of the various existing extraction protocols for musculoskeletal tissues. Here, we evaluated polar metabolite extraction from bone and muscle of mouse origin. The extraction methods compared were (1) modified Bligh–Dyer (mBD), (2) low chloroform (CHCl_3_)-modified Bligh–Dyer (mBD-low), and (3) modified Matyash (mMat). In particular, the central carbon metabolites (CCM) appear to be relevant for musculoskeletal regeneration, given their role in energy metabolism. However, the sensitivity, reproducibility, and robustness of these methods for detecting targeted polar CCM remains unknown. Overall, the extraction of metabolites using the mBD, mBD-low, and mMat methods appears sufficiently robust and reproducible for bone, with the mBD method slightly bettering the mBD-low and mMat methods. Furthermore, mBD, mBD-low, and mMat were sufficiently sensitive in detecting polar metabolites extracted from mouse muscle; however, they lacked repeatability. This study highlights the need for a re-thinking, towards a tissue-specific optimization of methods for metabolite extractions, ensuring sufficient sensitivity, repeatability, and robustness.

## 1. Introduction

Recently, there has been increasing interest in applying metabolomics to understand the pathophysiology of musculoskeletal disorders and the subsequent regeneration [1,2]. This stems from the fact that metabolism plays an integral role in maintaining physiology homeostasis when the body experiences an increased metabolic rate, loss of total body water, and increased collagen and cellular turnover, to supply the injured area with the appropriate cellular activity to re-build injured tissues and enable healing. 

To date, the sampling for metabolomics in many clinical indications such as musculoskeletal regeneration has predominantly focused on peripheral body fluids, such as serum, plasma, and urine [3,4]. Such an approach can give a system-level understanding of metabolic changes due to diseases, and degeneration or regenerative processes that affect the whole human body. Metabolomics on the level of tissue samples, e.g., bone, muscle, tendon, etc., is essential to understand the local healing and regeneration processes and tissue-level specific metabolic cascades directly relevant to the pathophysiology of diseased tissue. The extraction of metabolites has been relatively well-studied for muscle [5,6], but bone remains a relatively under-investigated material. The challenge of metabolomics of tissue samples is the unique matrix effect on the ionization potential of individual metabolites of interest. These matrix effects can be observed either as a loss in response (ion suppression) or an increase in response (ion enhancement), compared to measurement in a matrix-free environment [7].

Consequently, the quantification of metabolites using mass spectrometry (MS)-based metabolomics often results in more heterogeneity and variability for tissue samples, i.e., bone or muscle, than peripheral body fluids [8]. Therefore, the tissue samples for MS-based metabolomics must be handled and processed following an optimized workflow; from tissue handling (e.g., quenching of metabolism, storage) to processing (e.g., homogenization, extraction) [9]. The matrix components of muscle are considered relatively homogeneous compared to those of bone, which comprises different bone tissue structures (e.g., compact vs. spongy) and the marrow, and frequently varies across skeleton sites. Additionally, due to the mineralized nature of bone tissue, an appropriate technique should be applied to facilitate maximum access of the extraction solvent to the tissue, to obtain a homogenous solution. Hence, this study evaluated two different homogenization methods for the processing of bone tissues. 

Following tissue homogenization, metabolite extraction is a crucial step, mandating the reproducibility of results and the possible range of metabolites that can be detected. Generally, the choice of metabolite extraction depends on the analytical tools and the metabolites of interest. This study focused on the central carbon metabolites (CCM), the collection of products that result from the complex networks of reactions responsible for the transport and oxidation of the main sugars inside a cell. This include metabolic pathways such as the tricarboxylic acid cycle, pentose phosphate pathway, amino acids, glycolysis, etc. [10]. For the metabolite extraction, this study compared three different biphasic extraction methods modified from the procedure originally proposed by Bligh and Dyer [11] and Matyash et al. [12] for mouse bone and muscle. The Bligh–Dyer (BD) method was one of the first used in metabolomics to extract polar and non-polar fractions simultaneously. This study used two modified versions of the BD method, where the ratio of chloroform (CHCl_3_) was varied, and the modified Matyash (mMat) method, where CHCl_3_ was substituted with Methyl tert-butyl ether (MTBE), a chemical with better stability and less toxicity than CHCl_3_. Subsequently, efforts were aimed at comparing the repeatability and sensitivity of the methods, using gas chromatography-mass spectrometry to detect the CCM.

## 2. Results

The performance of the methods was compared by targeted analysis of 75 specific metabolites from the CCM. The following biological classes were covered: glycolysis, amino acids and metabolites from the tricarboxylic cycle, as well as some chemical classes, herein named as “others” (nucleobases/nucleosides, phosphate compounds, sugars, carboxylic acids and glycerol metabolites), as already demonstrated in previous studies [13,14].

Data reproducibility is usually reported as the relative standard deviation (RSD in %) of a repeated sample measurement in the metabolomics field. In the GC-MS metabolomics analysis, the maximum generally accepted tolerance of RSD is 30% for any individual metabolite [15]. An RSD of less than 10–15% is considered a good reproducibility [16]. 

### 2.1. Tissue Homogenization: Tissuelyzer vs. Pulverizer

Physical disruption of the tissue is a prerequisite for successful metabolite extraction [17], especially for bone (hard tissue). In this study, two homogenization methods, Tissuelyzer and Pulverizer, were compared for bone. The repeatability (represented by the median relative standard deviation (mRSD)) and sensitivity in metabolite quantification (represented by the number of metabolites detected) were assessed, as displayed in Table 1.

Bone samples processed using the Tissuelyzer yield an mRSD value of 31% (Table 1), varying between 2% (glycine) and 98% (cytosine) for individual metabolites. Of the 38 metabolites detected, 18 metabolites had RSDs below 30%, of which seven metabolites were below 15%. By comparison, bone samples processed with the Pulverizer yielded an mRSD value of 40%, ranging from 10% (dihydroxyacetone phosphate and lactic acid) to a maximum of 142% (proline) for individual compounds. Out of the 36 metabolites detected, 16 metabolites exhibited RSD values below 30%, with four that were below 15%.

This study used methoximation (MeOx) followed by trimethylsilylation (TMS) to derivatize the samples. Such derivatization can yield incomplete or isomeric derivative peaks. The repeatability of derivatives of TMS and MeOX products are listed in Table 2. It was also noted that adenine, cytosine, and isoleucine were only detected in bone samples homogenized with Tissuelyzer, while tyrosine was only detected in bone samples homogenized with Pulverizer. Generally, the derivatives showed better repeatability (lower RSD values) for samples processed with Tissuelyzer than Pulverizer.

### 2.2. Bone (Mice)

The mice bones (*n* = 5) were homogenized using the Tissuelyzer, and the polar metabolites were extracted following the mBD, mBD-low, or mMat method. All samples were analyzed as a single batch on the GC-MS. The three extraction methods were compared in terms of sensitivity (the number and intensity of metabolites detected), robustness (number of missing values), and repeatability (RSD distribution and mRSD). Although five replicates per class were initially prepared (total samples: 15), four samples were lost due to missed injections, leaving four per class for mBD and mBD-low, and three per class for mMat.

A Venn diagram (Figure 1a) illustrates that the metabolites detected from the bone differed for the three tested extraction methods. The mBD method yielded 65 metabolites, the mBD-low method resulted in 60 metabolites, and the mMat method detected 59 metabolites. Among the detected metabolites, 58 compounds were commonly detected in all samples, regardless of the extraction methods. Five metabolites were exclusively detected in mBD, of which two were amino acids (isoleucine and serine), one was a compound from the tricarboxylic acid cycle (2-oxoglutaric acid), and one was from the glycolysis pathway (6-phosphate glyceric acid).

The metabolites extracted from mouse bone using the mBD method yielded an mRSD of 15%, ranging from 1% (2-hydroxy glutaric acid and mannose) to 52% (D-pantothenic acid) for individual compounds (Figure 1b). Among the 65 detected metabolites, 24 metabolites showed RSDs of less than 10%, indicating very good reproducibility, 28 had RSDs between 10 to 30%, and 13 had RSDs above 30%.

Using the mBD-low method, the mRSD for all the 60 metabolites detected was 18, ranging from 3% (myoinositol) to 79% (ribose-5-phosphate) for individual compounds. Of which, 15 metabolites had RSDs of less than 10%, 29 metabolites with RSD between 10 and 30%, and 16 metabolites with RSDs >30%.

Metabolites extracted using the mMat method had a mRSD of 15%, with RSDs of individual compounds ranging from 1% (erythrose-4-phosphate, glucose and mannose) to 77% (pyruvic acid). Of the 59 metabolites detected, 24 metabolites showed RSDs of less than 10%, 27 metabolites showed RSDs between 10 and 30%, and eight metabolites had RSDs >30%.

The repeatability (as measured by the mRSD) of the metabolites based on biological classes was best for tricarboxylic acid cycle (4% for all three methods), followed by glycolysis (mBD: 20%, mBD-low: 26%; mMat: 24%) and amino acids (mBD: 25%, mBD-low: 18%, mMat: 23%).

The three methods were relatively comparable in terms of robustness, as the number of detected and missing metabolites were similar for all biological repeats, as illustrated in Table 3. The comparability between the number of metabolites detected for the three methods was assessed with a Kruskal–Wallis-test (alpha = 0.05), which resulted with a *p* = 0.016. The null-hypothesis was then rejected, and the number of metabolites detected between the three methods was considered significantly different.

Sensitivity was additionally assessed by comparing the relative intensities of individual metabolites across the three methods (Figure 2). For this, the log2 ratio (fold-change) of the mean peak areas of each metabolite was plotted with respect to mBD. This method was used as a baseline, because mBD is the normal method used in our lab [18]. Results greater than zero indicate that other methods produced larger peak areas (i.e., were more sensitive) for any given metabolite than mBD, and results below zero suggest the opposite. The three methods performed similarly, with mBD-low and mBD being close together for most of the metabolites, and mMat presenting values slightly lower than mBD. Specific metabolites, including erythrose-4-phosphate and ribulose-5-phosphate were notably better extracted using mBD-low or mMat.

### 2.3. Muscle (Mice)

Muscles harvested from mice (*n* = 5) were processed following the homogenization and polar metabolites extraction methods previously used for mouse bone. As illustrated in the Venn diagram (Figure 3a), 63 metabolites were detected in the muscle samples extracted with the mBD method. In comparison, only 59 and 57 metabolites were detected in muscle samples that underwent extraction using mBD-low and mMat, respectively. Among all the detected metabolites, 54 metabolites were simultaneously detected by all three extraction methods. 

The repeatability of the polar metabolites extracted from mouse muscle was evaluated based on the mRSD of the peak area of the detected metabolites. The metabolites extracted from mouse muscle using the mBD method yielded an mRSD of 35%, ranging from 3% (pyroglutamic acid) to 112% (erythrose-4-phosphate) for individual compounds. Among the 63 detected metabolites, two metabolites had an RSD of less than 10%, indicating very good reproducibility, 25 metabolites had RSDs between 10% and 30%, and 36 metabolites had RSDs >30%.

Using the mBD-low method gave an mRSD of 46% across the 59 metabolites detected, ranging from 7% (adenosine and lactic acid) to 88% (pyruvic acid) for individual compounds. Of which, two metabolites had RSDs of less than 10%, 17 metabolites had RSDs between 10% and 30%, and 40 metabolites had RSDs >30% (Figure 3b). 

Metabolites extracted from mouse muscle using the mMat method yielded an mRSD of 47%, with RSDs of individual compounds ranging from 8% (pyruvic acid) to 93% (ribitol). Of the 57 metabolites detected, two metabolites showed RSDs of less than 10%, 15 metabolites showed RSDs between 10 and 30%, and 40 metabolites had RSDs >30%. 

The repeatability (mRSD) of the metabolites based on biological classes, i.e., glycolysis (mBD: 43%, mBD-low: 65%, mMat: 43%) and amino acids (mBD: 45%, mBD-low: 53%, mMat: 54%), were above 30% for all extraction methods. The mRSD of metabolites of the tricarboxylic acid cycle (mBD: 26%, mBD-low: 56%; mMat: 42%) was only marginally lower than that of metabolites from the glycolysis or amino acids families.

The three methods were relatively comparable in terms of robustness, as the number of detected and missing metabolites was similar for all biological repeats, as illustrated in Table 4. The comparability between the three methods was assessed with a Kruskal–Wallis test (alpha = 0.05), which resulted in p= 0.008. The null-hypothesis was rejected, meaning that the number of metabolites detected between the three methods was significantly different.

The sensitivity for individual metabolites is illustrated in Figure 4. The log2 ratio (fold-change) of the mean peak areas of each metabolite was plotted. The method used as a baseline was mBD. Generally, a much bigger difference was observed between mBD vs. mBD-low and mMat, but especially for the latter. The mBD-low method performed rather poorly in detecting fumaric acid, 6-phosphogluconic acid, and ribose-5-phosphate. By comparison, the mMat method was significantly more sensitive to asparagine, lactic acid, and urea than the mBD method.

## 3. Discussion

Metabolomics is a powerful platform for high-throughput quantification of small molecule metabolites of biological specimens. Prior to the screening with GC-MS, acquired tissue specimens, i.e., bone or muscle, must undergo tissue homogenization, metabolite extraction, and derivatization. However, there is a lack of consensus on the optimum methods for extracting metabolites from tissue specimens, especially if different tissues such as bone and muscle are compared. 

The first problem to overcome with solid tissues, such as bone and muscle, is finding a way to measure technical replicates without distorting the extraction process. Here, biological replicates from in-bred healthy mice were used, with the assumption that the within-group biological differences between the mice would be less than the between-method technical differences between the extraction methods. Moreover, in this study, tissue specimens were harvested from male mice, thereby minimizing the biological variability by mitigating the influence of sex on the metabolome, as described by Caterino et al. [19]. Nonetheless, previous experiments on the same breed of mice suggest a median biological variability of 17%, although this can obviously vary greatly between individual metabolites. This allows for a certain experimental error in our results, which we acknowledge.

Homogenization of tissue is also a crucial step that allows for sufficient penetration of solvents through the tissue matrix to extract metabolites. Generally, homogenization is more challenging for hard tissue, i.e., bone, than soft tissue, i.e., muscle. A previous study indicated that the 24 Precellys homogenizers were optimal for the homogenization of muscle [20]. Here, two homogenization methods for mice bones were compared: the Tissuelyzer vs. the Pulverizer. The Tissuelyzer uses rapid agitation of beads to disrupt tissue and lyse cells in the presence of extraction solvent. By contrast, the Pulverizer uses a calibrated and controlled mechanical force to cryofracture flash-frozen tissues in the dry state and the extraction solvents are added to the crushed tissue afterwards. Overall, the Tissuelyzer slightly outperformed the Pulverizer for bone, evidenced by the higher number of detected metabolites and the lower mRSD. The results suggest that the presence of the extraction solvents during tissue homogenization may allow for better penetration of the solvent through the mineralized bone matrix, resulting in a better outcome during data acquisition with GC-MS. 

The suitability of the three extraction methods (mBD, mBD-low, and mMat) for extracting metabolites from mouse bone and muscle was assessed in terms of sensitivity, robustness, and repeatability for targeted CCM profiling. Extraction methods were based on published literature [13,14,15]. As expected, the selection of different extraction solvents and ratios impacted the number of metabolites detected and the RSD values. 

The mBD method was more sensitive compared to the mBD-low and mMat methods, as evidenced by the higher number of metabolites detected (fewer missing metabolites) in both bone and muscle samples. This phenomenon was most likely due to the greater volume of CHCl_3_ present in mBD compared to mBD-low, enabling full extraction of the lipids into the lipid phase and resulting in better detection of polar metabolites, with less contamination of non-polar compounds. Similarly, Nam et al. [21] reported global profiling of mouse bone tissue using MeOH: CHCl_3_: H_2_O (*v*/*v*/*v* ratio of 1:2:1.5), and both polar and non-polar metabolites achieved a high repeatability and extraction yield (as measured by the RSDs in their QC samples). In agreement with that, and supporting our findings, different studies using muscle tissue [22,23], reported that MeOH:CHCl_3_:H_2_O combinations with the *v*/*v*/*v* ratios of 1:1:0.5 and 1:1:0.8, respectively, obtained a high metabolite yield, proving its suitability for extraction of targeted polar metabolites.

The mBD and mMat methods showed similar reproducibility for extracting the tested polar metabolites from mouse bone, with the mRSD value of 15%, while mBD-low showed slightly lower reproducibility, with a mRSD value of 18%. These results are consistent with previous observations, where polar metabolites were extracted from tissue, obtaining mRSD values ranging from 2% to 25% [24]. Conversely, the polar metabolite extraction from mouse muscle showed poor reproducibility, with the mRSD value of 35% for mBD, 46% for mBD-low, and 47% for mMat. It is known that the absolute concentration of a metabolite and the amount of matrix in a sample affect the metabolite’s ionization performance [25]. Hence, either the different matrix compositions between bone and muscle or the analyzed concentrations could be the underlying cause of the considerable difference in reproducibility of polar metabolite extraction. In this study, the tissue to solvent ratio of 50: 1 (mg/mL) was used for bone and muscle, which might not be optimal for muscle samples. Indeed, Zukunft et al. [26] pointed out that using a high tissue to solvent ratio (1:3 and 1:6 mg/µL) for muscle resulted in reproducible detection of metabolites. We also cannot rule out that the high RSD values may reflect very real tissue to tissue biological differences in muscle, given that it is a very metabolically active tissue. 

When looking at the different metabolite classes separately, the reproducibility results were similar for mBD and mBD-low. However, for mMat, the RSD values of the different metabolite groups were generally higher for both tissue types. It is not clear from the experiments conducted here whether this was due to the physico-chemical effects of MTBE as a solvent resulting in a less reproducible extraction, the volume or ratio not being optimum, or the inadvertent contamination of the polar phase with organic compounds on recovery, because the polar and inorganic phase layers were reversed in order. Numerous studies have investigated the sample extraction efficiencies of biphasic methods when extracting polar metabolites from the aqueous layer [27,28,29]. From a chemistry perspective, the differences in reproducibility could be attributed to the solubility index of each metabolite in the solvents used. In this study, the polar solvents were the same for all methods (MeOH and H_2_O), but the different non-polar phase solvents (CHCl_3_ or MTBE) used may influence which lipid species stay in the polar phase or cross into the non-polar phase. The polarity index of MTBE was 2.5, compared to chloroform’s 4.1 [30]. Previous studies reported variability in CHCl_3_ and MTBE for lipid extraction, which may affect the final composition, matrix, and derivatization effects and, thus, the reproducibility of the polar phase results [12,29,31]. Another factor contributing to the observed differences in reproducibility between the three tested methods is the absolute volumes and ratios of each polar and non-polar solvent. Too small a volume of either polar or non-polar phase, and the extraction is not completed, due to the metabolites overwhelming the solvent capacities [32]. The high CHCl_3_ volume ratio in the mBD method is likely to have contributed to complete phase separation and efficient extraction of polar metabolites, improving the repeatability. The mBD-low method had the lowest non-polar solvent volume and the highest polar solvent volume, which has the dual effects of more solvent capability and increased dilution of the final extract (less is sometimes more with mass spectrometry), contributing to the relatively good repeatability and sensitivity.

Overall, the extraction of polar metabolites using any of the methods, i.e., mBD, mBD-low and mMat, has sufficient robustness and repeatability for bone. A high number of the metabolites detected are included in the internationally accepted criterion of a RSD below 15% for bioanalytical method validation [33]. The mBD method performed slightly better than mBD-low and mMat in sensitivity and reproducibility for detecting polar metabolites extracted from mouse bone. Our results suggest that mBD, mBD-low, and mMat are sufficiently sensitive in detecting polar metabolites extracted from mouse muscle; however, all three methods require improvement in their repeatability measures, and larger solvent to tissue volumes may be required, or true technical replicates for better assessment.

## 4. Materials and Methods

### 4.1. Sample Acquisition and Storage

Two types of biospecimens were collected: (1) bone (femur) and (2) muscle (biceps femoris) from mice (C57Bl/6J, male, 12 weeks old, ethics approval X9007/17). Mice were killed by cervical dislocation and samples were collected (both femurs from each animal), immediately flash-frozen in liquid nitrogen, and stored at −80 °C. All animals were fed the standard diet comprised of 53% carbohydrate, 11% fat, and 36% protein (product code: V1124 from Spezialdiäten GmbH, Soest, Germany).

### 4.2. Sample Preparations

Two principle experiments were carried out during the development of this analysis protocol.

Experiment 1 (described in Section 4.2.1): A comparison of two different homogenization processes of samples was performed for bone, only due to the hardness of the tissue. Here, two different methods were tested and compared (Tissuelyzer vs. Pulverizer). The metabolites were extracted using the mBD-low protocol previously established in the lab [18].

Experiment 2 (described in Section 4.2.2): Three different metabolites extraction methods were tested and compared for bone and muscle harvested from the same mice. All tissues were homogenized with Tissuelyzer, based on the outcome of experiment 1. For all the experiments, five biological replicates (*n*= 5) were prepared for each homogenization/extraction condition.

#### 4.2.1. Tissue Homogenization

Experiment 1—Homogenization method 1: Samples (bone) were lysed using a Precellys Fast Prep24 Tissue Homogenizer (Bertin Instruments, Montigny-le-Bretonneux, France). Between 30 and 60 mg of tissue was placed in a 2-mL tube (Precellys^®^ 24, Bertin Instruments). Then, methanol (MeOH) and CHCl_3_ solution pre-cooled at 4 °C were added, using a solvent-tissue ratio of 1 mL/50 mg. Metallic beads (lysing matrix SS 116921050-CF, MP Biomedicals) were added to the tubes. Samples were lysed with a cycle of 3 × 30 s bursts at 6 m/s, with a break of 5 s between cycles.

Experiment 1—Homogenization method 2: An automated pulverizer (CP02 cryoPREP^®^ Covaris) was used. Briefly, the sample was inserted in an appropriate pulverizer tube (tissueTube TT1, Covaris, Brighton, UK), immersed in liquid nitrogen, and placed in the pulverizer to be crushed while in a frozen state. The pulverized contents were then transferred to a new Eppendorf vial, where the subsequent metabolic extraction was then performed.

#### 4.2.2. Extraction of polar metabolites

Experiment 2—Three different extraction methods were tested and compared for bone and muscle harvested from mice. The Bligh–Dyer (BD) and the Matyash protocol [34] are biphasic extraction methods commonly used to obtain polar and non-polar metabolites from serum or plasma. The modified versions of the BD and Matyash methods previously established [18,34,35] were tested in this study.

For each method and tissue type, i.e., bone or muscle, *n* = 5 biological replicates were used. After extraction, 300 µL of the polar phase was dried overnight at 30 °C at a speed of 1550× *g* at 0.1 mbar using a rotational vacuum concentrator (RVC 2–33 CDplus, Christ, Osterode am Harz, Germany). All samples were stored dry at −80 °C, until further processing. To measure the technical variability of the instrument, representative quality control (QC) samples were prepared by pooling the sample extracts after each extraction method.

Extraction method (1) modified Bligh-Dyer (mBD) [35]: First, 1 mL of MeOH: water (H_2_O) (1:1, *v*/*v*) mixture was added to 50 mg tissue, following the addition of CHCl_3_, maintaining a 1:1:1 (*v*/*v*/*v*) ratio between solvents. Then, 2 µg/mL of cinnamic acid internal standard was added to the extraction solvent. Samples were shaken at 4 °C at 800 rpm for 45 min and centrifuged at 4700× *g* at 4 °C.

Extraction method (2) modified Bligh–Dyer method with low CHCl_3_ (mBD-low) [18]: This method followed the mBD method with a modification on the solvent ratios from 1:1:1 MeOH:CHCl_3_:H_2_O (*v*/*v*/*v*) to 1: 0.4: 1. This method has been used previously in other studies and has the advantage that it uses less chloroform. However, the lower volume of chloroform can lead to unreliable extraction for samples with a high lipid content.

Extraction method (3) modified Matyash (mMat) [34]: Stepwise, this method is similar to the mBD, but the solvents and solvent ratios were different. For this method, instead of CHCl_3_, MTBE was used. The final ratio of solvents MeOH: MTBE: H_2_O was 1:1.3:1.2 (*v*/*v*/*v*). Samples were homogenized in 75% cold MeOH. Then, MTBE was added. Samples were then shaken at 1000 rpm for 3 min. Finally, H_2_O was added together with 2 µg/mL of cinnamic acid internal standard. Samples were shaken at 1000 rpm for 1 min and centrifuged for 10 min at 2415× *g* at 18 °C. 

#### 4.2.3. Sample Derivatization

Prior to derivatization, frozen samples were dried in a rotational vacuum concentrator (RVC 2–33 CDplus, Christ, Osterode am Harz, Germany) for 60 min to remove any residual water. Subsequently, the dried extracts were dissolved in 15 µL of methoxyamine hydrochloride solution (40 mg/mL in pyridine) and incubated for 90 min at 30 °C at 800 rpm. Then, 50 µL of N-methyl-N-[trimethylsilyl]trifluoroacetamide (MSTFA) was added to the solution, incubated at 37 °C for 60 min and centrifuged for 10 min at 18,213× *g*. Finally, 25 µL aliquots were prepared in glass vials and closed with appropriate lids (Labconco, Kansa City, MI, USA) for GC-MS measurements.

An identification mixture was prepared and derivatized using the same method, to ensure reliable compound identification. Additionally, an alkane mixture for a reliable retention index calculation was included [35]. Pooled QC samples were treated in the same way.

### 4.3. Data Acquisition by Gas Chromatography-Mass Spectrometry (GC-MS)

The GC-MS analysis of CCM was performed on a Pegasus 4D GCxGC TOFMS-System (LECO Corporation, St. Joseph, MN, USA) complemented with an auto-sampler (Gerstel MPS DualHead with CAS4 injector, Mühlheim an der Ruhr, Germany). The samples were injected in split mode (split 1:5, injection volume 1 µL) in a temperature-controlled injector with a baffled glass liner (Gerstel, Mühlheim an der Ruhr, Germany). The following temperature program was applied during sample injection: 2 min to allow for column equilibration at 68 °C, followed by a serial increment of temperature, from 68 °C to 120 °C (5 °C/minute), from 120 °C to 200 °C (7 °C/minute), and from 200 °C to 320 °C (12°C/minute). The maximum temperature, 320 °C, was maintained for 7.5 min. 

Gas chromatographic separation was performed in one-dimensional mode on an Agilent 7890 (Agilent Technologies, Santa Clara, CA, USA), equipped with a VF-5ms column (Agilent Technologies, Santa Clara, CA, USA) of 30 m length, 250 µm inner diameter, and 0.25 µm film thickness. Helium was used as the carrier gas, with a flow rate of 1.2 mL/minutes. Spectra were recorded in a mass range of 60 to 600 *m*/*z* with 10 spectra/second. 

All samples were run in scan mode, to obtain targeted and untargeted metabolic profiles. We pre-selected 75 key CCM to target for data analysis (Appendix A), based on previous work in our lab [14].

### 4.4. Data Processing and Analysis

The GC-MS chromatograms were processed with ChromaTOF software (LECO Corporation, St. Joseph, MN, USA) including baseline assessment, peak picking, and area computation.

Once the pre-processing of GC-MS data was completed, the data were exported and merged with an in-house written R script. A total of 5 biological replicates were prepared for each method. A minimum of 3 valid values per metabolite had to be present to consider the metabolite as present. The peak area of each metabolite was calculated by normalization to the internal standard, cinnamic acid. Relative quantities were used, and derivatives from the same original metabolite species were summed up for the final data analysis. 

A Kruskal–Wallis test was performed using IBM SPSS Statistics 27.0 for statistical analysis. R studio (Version 1.3.1056) software was used for data analysis purposes. The analyzed metrics included the number of compounds, missing values, and the mRSDs of individual metabolite and biological classes.

## 5. Conclusions

Generally, this study highlights the relevance of the careful preparation of tissue sampling and processing, specifically if such diverse tissues as bone and muscle are to be processed before GC-MS. In the first step of tissue processing, the results indicated that Tissuelyzer yielded a better homogenate of mouse bone for extracting polar metabolites than Pulverizer. While extracting polar metabolites from bone or muscle, the types and ratio of the solvent mixture and the tissue to solvent ratio can impact the sensitivity, robustness, and reproducibility of the GC-MS profiling of CCM.

## Figures and Tables

**Figure 1 metabolites-12-00453-f001:**
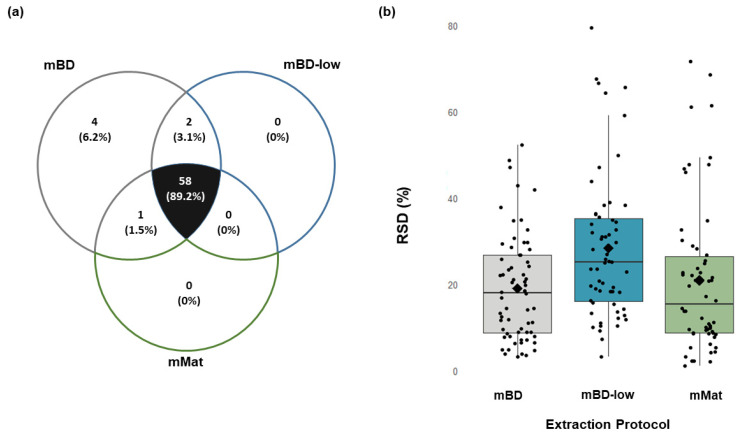
Influence of the different extraction solvents in mouse bone tissue. (**a**) Venn diagram with principal central carbon metabolites detected (and relative percentages) between the three extraction methods, i.e., modified Bligh–Dyer (mBD), modified Bligh–Dyer with low chloroform (mBD-low), and modified Matyash (mMat). (**b**) Distribution of individual metabolites’ relative standard deviation (RSD) for the different extraction methods. Each black point represents an RSD for a single metabolite. The 58 metabolites common to all three methods are shown here, and the median relative standard deviation (mRSD) of these is represented by the black middle line of the boxplot.

**Figure 2 metabolites-12-00453-f002:**
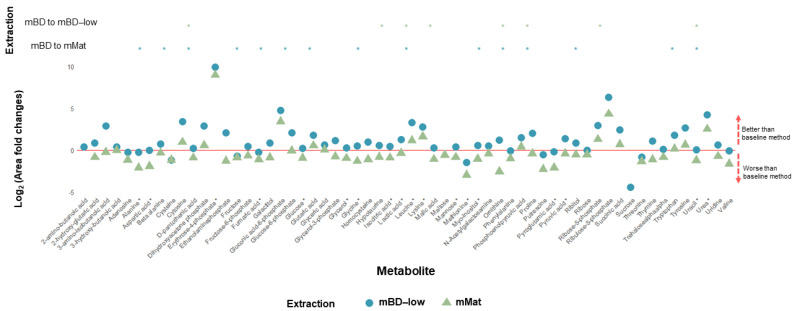
Bone tissue: Relative quantification represented as fold change of single metabolite peak areas using cinnamic acid normalization. Modified Bligh–Dyer (mBD) was used as a baseline (represented by the red line). Extraction sensitivity with significant variation represented in the upper pane. *: *p*–value < 0.05.

**Figure 3 metabolites-12-00453-f003:**
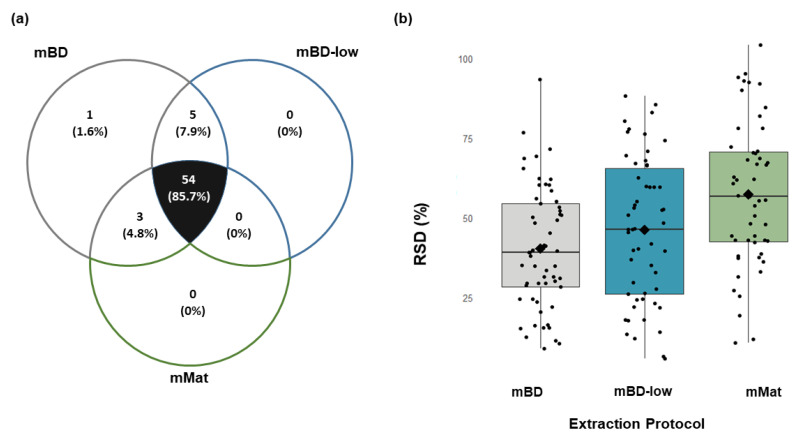
Influence of the different extraction solvents in mouse muscle tissue. (**a**) Venn diagram with principal central carbon metabolites detected (and relative percentages) between the three extraction methods, i.e., modified Bligh–Dyer (mBD), modified Bligh–Dyer with low chloroform (mBD-low), and modified Matyash (mMat). (**b**) Distribution of individual metabolites’ relative standard deviations (RSD) for the different extraction methods. Each black point represents an RSD for a single metabolite. The 54 metabolites common to all three methods are shown here, and the median relative standard deviation (mRSD) is represented by the black middle line of the boxplot.

**Figure 4 metabolites-12-00453-f004:**
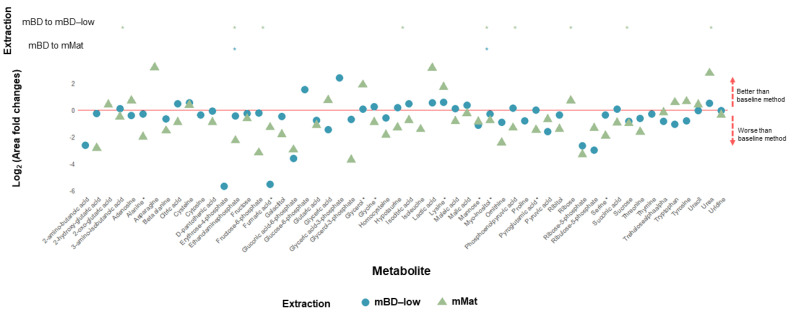
Muscle tissue: Relative quantification represented as fold change of single metabolite peak areas using cinnamic acid normalization. Modified Bligh–Dyer (mBD) used as baseline and represented by the red line. Extraction sensitivity with significant variation represented in the upper panel. *: *p*–value < 0.05.

**Table 1 metabolites-12-00453-t001:** Comparison of repeatability (median standard deviation = mRSD) and sensitivity (number of metabolites detected) in mouse bone samples using the Tissuelyzer or Pulverizer.

Homogenization Method	No. Biological Replicates	No. Metabolites Detected	mRSD (%)
Tissuelyzer	5	38	31 ± 5
Pulverizer	5	36	40 ± 5

**Table 2 metabolites-12-00453-t002:** Analysis of the repeatability of derivatives of Trimethylsilylation (TMS) and Methoximation (MeOX) products.

Homogenization Method	Derivatives	Trimethylsilylation (TMS)				Methoximation (MeOX)
		4-TMS	3-TMS	2-TMS	1-TMS	
Tissuelyzer	No of metabolites	5	12	17	4	8
	RSD range (%)	11 to 75	5 to 107	8 to 98	5 to 38	6 to74
	mRSD (%)	24	28	31	16	36
Pulverizer	No of metabolites	5	13	13	4	8
	RSD range (%)	29 to 164	12 to 86	10 to 142	13 to 82	10 to 67
	mRSD (%)	50	41	38	60	38

**Table 3 metabolites-12-00453-t003:** Summary of the number of metabolites detected and missing from the mouse bone after performing extraction. The metabolites were extracted with the modified Bligh–Dyer (mBD), mBD with low chloroform (mBD-low), or modified Matyash (mMat) method. mRSD: median relative standard deviation in %, NA: no value available.

	mRSD (%)	No. Metabolites Detected	Missing Metabolites	Mean No. NA’s for Each Sample (%)	No. Replicates Used for Analysis/Total Prepared
mBD	15	65 ± 2	1 ± 2	6 ± 1	4/5
mBD-low	18	60 ± 0	6 ± 0	11 ± 1	4/5
mMat	15	59 ± 0	7 ± 0	11 ± 0	3/5

**Table 4 metabolites-12-00453-t004:** Summary of the number of metabolites detected and missing from mouse muscle after extraction. The metabolites were extracted with the modified Bligh–Dyer (mBD), mBD with low chloroform (mBD-low), or modified Matyash (mMat) methods. In case of mBD, a total of four replicates were used for analysis purposes, due to a misinjection in one of the samples initially prepared. mRSD: median standard deviation in %. NA: no value available.

	mRSD (%)	No. Metabolites Detected	Missing Metabolites	Mean No. NA’s for Each Sample (%)	No. Replicates Used for Analysis/Total Prepared
mBD	35	63 ± 2	3 ± 2	4 ± 2	4/5
mBD-low	46	59 ± 1	7 ± 1	9 ± 1	5/5
mMat	47	57 ± 2	9 ± 2	11 ± 4	5/5

## Data Availability

The data presented in this study are available in article and Appendix A.

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
