# Peer review of "A Comparison of Solvent-Based Extraction Methods to Assess the Central Carbon Metabolites in Mouse Bone and Muscle"

_metabolites, 2022, doi:10.3390/metabo12050453_

Round 1

Reviewer 1 Report

This paper would lend itself much better to the more recent format of a communication.  There is not much difference in how the paper would be formated and I recommend the approach to the authors as very appropriate. 

The paper is not in Metabolites format, witht the materials and methods at the end of the paper.  This should be adjusted and the materials and methods should be more clear about the N involved.  If 5 mice were used, state that clearly. 

There are some minor english issues, plurals for singular etc and the writing is a bit obtuse with the excessive use of abbreviations where words woud be fine. 

Try to be less obtuse  with the language and lead your reader gently through the essentials of the paper.

Author Response

Dear Reviewer 1,

Thank you for your comments, please find attached our response.

Best,

Reviewer 2 Report

The authors compared different sample homogenization methods and metabolite extraction methods in extracting polar compounds, particularly the polar central carbon metabolites (CCM), from mouse bone and muscle for GCMS analysis. The extraction methods were (1) modified Bligh-Dyer (mBD), (2) low chloroform (CHCl3)-modified Bligh-Dyer (mBD-low), and (3) modified Matyash (mMat). The targeted metabolites are 75 glycolysis, TCA, and PPP pathway metabolites. All these methods were found to have sufficient robustness and repeatability, with mBD performs slightly better in terms of number of metabolites detected, median RSD, and missing metabolite number.

The manuscript was well written. I only have a minor comment. It is very helpful if RSD is given for each detected metabolites in the Supplementary Information so that people can gauge the repeatability of each compound.

Author Response

Dear Reviewer,

Thank you for the comment made. Please find attcahed our answer.

Best,

Daniela

Author Response

Dear Reviewer,

We thank you for the valuable comments made. Please find attached our answers.

Best,

Daniela

Round 2

Reviewer 1 Report

I previouslly suggested this be converted to a communication rather than article considering the scope of the new information provided and the advantage of a succinct presentation.  I again encourage that consideration.  The paper is much longer that needed to present the findings, which should not have been unexpected considering what has alreadly been published in the literature.  Basically the work confirms the rough estimation that sample preparations involving extractions have roughly 15% or more error involved in techniques used in the process. 

There are two english usage issues I have marked, one involving the use of the plural of mouse in appropriately throughout.  This is indeed an oddity of the language but one very apparent to native speakers.  The other is the common misusage of the word since as a substitute for because.  Since is only valid in reference to time frame.

The assignment of unconventional abbreviations to terms seriously damages the readability of a paper.  The authors should instead use the intact term, but perhaps not use them so repeatitively.  This paper is overly long and could be better crafted in that regard.  An example would be the discussion which is over 1700 words and tries to address possibilities that are very unlikely. 

In the methods improvements are made but it is not clear if the mice were dead or anesthetized when sampled, nor how which ever was accomplished.  It is not clear if both femurs were taken or one.  The actual source of the feed was not specified though this reviewer will admit the feed fed is likely not particularly relevant to the findings at the level reported. 

The statistical approach is reasonable but again perhaps the use of RSD when you are simply reporting summary statistics (standard deviations) is a bit pompous.  Everything in those sections could be stated much more succinctly. 

Author Response

Dear Reviewer,

We thank you for the comments. Attached find our answers.

Best,

Reviewer 3 Report

I am satisfied with the responses provided by the authors. Thanks for the clarifications. The article can now be accepted in the present form.

Author Response

Dear Reviewer,

Thanks for taking the time to review our article.

Best,